# Antifreeze Proteins: Novel Applications and Navigation towards Their Clinical Application in Cryobanking

**DOI:** 10.3390/ijms23052639

**Published:** 2022-02-27

**Authors:** Marlene Davis Ekpo, Jingxian Xie, Yuying Hu, Xiangjian Liu, Fenglin Liu, Jia Xiang, Rui Zhao, Bo Wang, Songwen Tan

**Affiliations:** Xiangya School of Pharmaceutical Sciences, Central South University, Changsha 410013, China; 217208001@csu.edu.cn (M.D.E.); xiaoxian.xie@csu.edu.cn (J.X.); yuyinghu@126.com (Y.H.); liuxiangjiancsu@163.com (X.L.); liufl1012@gmail.com (F.L.); 197211040@csu.edu.cn (J.X.); zr2498176546@163.com (R.Z.); 197211030@csu.edu.cn (B.W.)

**Keywords:** cryopreservation, antifreeze proteins (AFPs), antifreeze glycoproteins (AFGPs), cryoprotectants, ice recrystallisation (IR), thermal hysteresis (TH)

## Abstract

Antifreeze proteins (AFPs) or thermal hysteresis (TH) proteins are biomolecular gifts of nature to sustain life in extremely cold environments. This family of peptides, glycopeptides and proteins produced by diverse organisms including bacteria, yeast, insects and fish act by non-colligatively depressing the freezing temperature of the water below its melting point in a process termed thermal hysteresis which is then responsible for ice crystal equilibrium and inhibition of ice recrystallisation; the major cause of cell dehydration, membrane rupture and subsequent cryodamage. Scientists on the other hand have been exploring various substances as cryoprotectants. Some of the cryoprotectants in use include trehalose, dimethyl sulfoxide (DMSO), ethylene glycol (EG), sucrose, propylene glycol (PG) and glycerol but their extensive application is limited mostly by toxicity, thus fueling the quest for better cryoprotectants. Hence, extracting or synthesizing antifreeze protein and testing their cryoprotective activity has become a popular topic among researchers. Research concerning AFPs encompasses lots of effort ranging from understanding their sources and mechanism of action, extraction and purification/synthesis to structural elucidation with the aim of achieving better outcomes in cryopreservation. This review explores the potential clinical application of AFPs in the cryopreservation of different cells, tissues and organs. Here, we discuss novel approaches, identify research gaps and propose future research directions in the application of AFPs based on recent studies with the aim of achieving successful clinical and commercial use of AFPs in the future.

## 1. Introduction

Long before the term cryopreservation was coined, storage at lower or subzero temperatures has been used for simpler applications like food preservation [1]. With research and the evolution of fields such as cryobiology, we have become aware of the vast applications of cryopreservation and the many prospects within for innovation. Conventionally, improper storage of biological samples would cause exposure to microbial contamination [2] and storage at subzero temperatures in commonly used liquid nitrogen or dry ice would bring about loss of sample viability through cessation of metabolic and biochemical processes [3]. These disadvantages are primarily why cryopreservation is crucial and relevant in diverse applications like cell cultures, biobanking and biopreservation, in vitro fertilization (IVF), organ transplantation and artificial insemination, etc. [4,5]. Cryopreservation entails quite more than freezing and thawing of samples with retained viability because for each stage of the process enormous time, expertise and resources have been invested in research to determine the optimal freezing rate and medium for respective samples. 

The stages in cryopreservation generally involve the selection or harvesting of the sample, freezing, storage in liquid nitrogen and thawing but cryopreservation research has revealed some challenges majorly attributed to ice-induced injury or damage to cells during ice formation, ice nucleation and/or ice recrystallization (IR). 

To overcome these drawbacks, the addition of cryoprotectants or cryoprotective agents (CPAs) before sample freezing or vitrification has become the mainstay [6]. CPAs are categorized as permeable (dimethyl sulfoxide (DMSO), proline, glycerol, ethylene glycol (EG), propylene glycol (PG), etc.) or impermeable (polyvinyl alcohol (PVA), polyampholytes, trehalose, sucrose, graphene oxide, etc.) based on whether they can cross cell membranes [7]. There is a limit to which CPAs can be used owing majorly to toxicity and adverse reactions [8]. There is therefore a growing need for more biocompatible cryoprotectants [9], especially in biomedical sciences where they are needed to preserve complex and sensitive biomolecules ranging from organelles, cells, tissues and more complicated organs for diverse applications in fertility/reproductive health [10,11], organ transplantations (liver, lungs, heart, bone marrow) [12,13], treatment of chronic diseases like cancer and diabetes and to minimize the use of live animal models in the laboratory. This need has led scientists to explore antifreeze proteins as a potential cryoprotectant. AFP was discovered first in 1957 in arctic fish [14] (Figure 1), but their application is still quite limited especially clinically when compared to the other cryoprotectants. 

AFPs are impermeable CPAs which have also been referred to as thermal hysteresis proteins and ice structuring or binding proteins based on their mechanism of action (Figure 2) which is in their ability to reduce or inhibit ice crystal growth by binding to ice surfaces [15]. This binding activity inhibits IR by causing the freezing temperature to be suppressed below the melting point in a non-colligative manner in a process termed large thermal hysteresis [16]. AFP binding also results in melting hysteresis where the melting temperature is raised above the melting point. The combination of these processes is termed thermal hysteresis (TH), without which organisms would not thrive in frigid environments [16]. The interaction between AFPs and water and ice crystals at low temperatures to inhibit ice formation and growth has been studied and confirmed with diverse methodologies including quantum filtering-spin exchange nuclear magnetic resonance (NMR) [17] microfluidic experimentation [18], NMR microimaging and the more recent site-directed spin labelling performed by monitoring the activity of spin-labelled AFPs at low temperatures with electron paramagnetic resonance (EPR) and cryo-photo microscopy [19,20].

This review delves in-depth into the advances in applications of AFPs in hypothermic storage and cryopreservation. We have studied, documented and discussed with evidence the relevance of antifreeze proteins to cryopreservation of oocytes, sperm cells, embryos, cancer cells, various tissues and organs, foods, etc. The challenges encountered therein and approaches for future research are also explored, which will serve as a guide to the recent trends in AFP application and stir the intellect of researchers towards innovative thinking as we strive to attain clinical approved and commercial AFP usage in cryopreservation protocols. 

## 2. Classification and Chemistry of Some Commonly Used AFPs 

Presented below (Table 1) is a summary of the sources and chemical properties of AFPs available from the literature. Thermal hysteresis activity (THA) is a kinetic model reflective of the degree to which an AFP inhibits ice formation; hence the most important property to consider. It is therefore not surprising that there is a strong relationship between the thermal hysteresis activity of AFPs and the frequency of their application in research studies. For instance, moderate to hyperactive AFPs like *Flavobacterium frigoris* ice-binding protein (FfIBP), type III AFP, *Leucosporidium* ice-binding protein (LeIBP) and *Tenebrio molitor* AFP have been more extensively studied. TH is mostly determined using differential scanning calorimetry (DSC) [21] and AFPs are classified as hyperactive, moderate or low based on their adsorbent gap at the ice surface, adsorption pattern and plane specific binding. Hyperactive AFPs have the least adsorbent gap and can absorb onto multiple crystal planes [22]. Ortiz et al. has proposed using the fractional statistics theory of adsorption that TH could be dependent on the protein shape, size and number of active ice-binding domains [23].

## 3. Applications of AFPs in Cryopreservation

### 3.1. Oocytes and Embryos

Recently, numerous research has been performed to evaluate the substitutability of AFPs in a bid to achieve better cryopreservation of oocytes and embryos. Cryopreservation of oocytes is a major breakthrough in reproductive medicine, especially considering the ethical, religious or legal controversies surrounding embryo preservation in some regions [63]. Assays including intracellular adenosine triphosphate (ATP) concentrations, meiotic spindle and chromosome deformity evaluation, blastocyst cell count, apoptosis assays, presence of reactive oxygen species, mitochondrial activity and deoxyribonucleic acid (DNA) double-strand breaks (DSBs) are among those commonly used for evaluation. A study by Jo et al. [64] accounts for the first attempt to use type III AFP in the vitrification of in vivo matured mouse oocytes where it supported strongly the cryoprotective effects of AFP III based on the higher survival rate of the cells following vitrification and warming, and lower extent of developmental abnormalities after in vitro fertilization. Similar favourable results were obtained in the vitrification of immature murine oocytes with AFP III [65,66] and bovine oocytes with AFGP-8 [67] and *Glaciozyma* sp. ice-binding protein (LeIBP) [68]. 

To further the research, Lee et al. [69] performed a comparative study on the effect of AFPs on murine oocytes vitrification where the performance of three different AFPS was evaluated. The AFPs used in the study included Type III AFP, FfIBP and LeIBP. Results obtained from the study also confirmed the ability of AFPs to maintain the anatomical and physiological integrity of vitrified mouse oocytes. Treatment with AFPs was also found to improve embryo development following IVF of the cells post-cryopreservation as evidenced by better Cleavage rate, blastocyst rate and Blastomere count. The treated cells also possessed less apoptotic blastomere count depicting the anti-apoptotic effect of AFPs; the FfIBP and LeIBP treated groups had significantly reduced DNA DSBs than the control group. Treatment with AFP assisted chromosome stability as it improved the meiotic spindle organization recovery and chromosome alignment during the vitrification and warming with FfIBP being the most advantageous in this regard. Significantly reduced generation of reactive free radicals was also recorded with the AFP treated group of murine oocytes. 

Furthermore, embryo cryopreservation and the quality of embryos produced with cryopreserved gametes is of utmost importance, especially in clinical applications. Li et al. included *Anatolica polita*-derived AFP to the controlled slow freezing versus vitrification Sheep embryos where vitrification resulted in higher survival and embryo hatching rates, and 10 μg/mL of the AFP promoted hatching of slow-growing vitrified embryos [70]. Similarly successful applications include AFP III in rabbit [71] and turbot (*Scophthalmus maximus*) [72] embryos, AFP I and III in zebrafish embryo [73,74], AFP I in seabream (*Sparus aurata*) [75] and sheep embryos [76], LeIBP in bovine embryos [68]. To the best of our knowledge, no research evidence that supports the use of AFP in human cryobanking of human oocytes or embryos has been reported at the time of this review. 

Researchers have resorted to investigating other potential cryoprotectants like ectoine [77] since the use of conventional cryoprotectants like DMSO and glycerol is clouded with concerns of toxicity, and loss of viability [77]. Bearing the good prospects of AFP use in mind, continuous research in this direction will avoid setbacks like that observed in the cryopreservation of bovine oocyte cryopreservation with AFP III [78] where mitotic spindle integrity was not preserved and they might just be the solution to clinically applicable oocytes and embryo cryopreservation [79].

### 3.2. Sperm Cells

Effective long-term spermatozoa storage is crucial in reproductive medicine, especially for bioconservation, artificial insemination/in vitro fertilization and to accommodate the high demand of various animals and seafood. For instance, consumption of abalone is considered beneficial to health and of high demand in countries like China and Japan [80]; human assisted reproduction is therefore used to accommodate this demand. Sperm cryopreservation would be termed effective when sperm motility, post-thaw viability, integrity of DNA, acrosome, plasma membrane and mitochondrial membrane and fertility rate of the sperm cells are preserved. Achieving success in sperm cryopreservation is often hindered by threats like thermal shock, ice crystallization/recrystallization and osmotic stress. Leakage of enzymes [81] and reduced penetration of oocytes in glycerol cryopreserved spermatozoa [82,83] have been recorded post-thaw. 

Hossen et al. obtained better outcomes by supplementing cryopreservation of *Haliotis discus hannai* (pacific abalone) spermatozoa with AFP III [84]. AFPs including AFP I, AFP III, DAFP (*Dendroides canadensis* antifreeze protein) and AFGPs have been applied in sperm banking research, e.g. sea bream sperm [85,86], buffalo bull (*Bubalus bubalis*) sperm [87,88,89,90], ram sperm [91], rabbit sperm [71], Persian sturgeon (*Acipenser persicus*) sperm [92], rooster sperm [93], common carp (*Cyprinus carpio*) sperm [94], human sperm [95] and crab-eating macaque (*Macaca fascicularis*) sperm [96], where later research has provided more insight into the possible mechanism of action of AFP III [97]. AFP III improved survival by preserving protein expression and modulating the release of cytochrome C and free radicals at the molecular level. Jang et al. [98] compared LeIBP with AFP III in the preservation of Korean bull semen and LeIBP showed better outcomes even in embryo formation after IVF. 

Contrary to the above studies, Xin et al. report no significant improvement using AFP I and III in the cryopreservation of sterlet (*Acipenser ruthenus*) sperm [99] and no protection against apoptosis in mouse sperm cells [100]. This might be suggestive of the need for further research into optimal conditions for sperm preservation where the effect of higher AFP concentrations and other types of AFPs can be evaluated. 

### 3.3. Cancer Cells, Stem Cells and Organoids

Cryobanking of cancer and stem cells have become an integral part of cell-based therapy [101] where the prolonged storage of these cells can allow for better research, understanding and treatment of complicated diseases such as cancers, autoimmune disorders, etc. [102,103]. A potent approach to diminish the toxic effect of DMSO in the cryopreservation of cancer and stem cells has been a combination with other CPAs like trehalose [104], sucrose [105], polyampholytes [106] and AFPs. This creates a synergy in ice inhibiting mechanisms while reducing the total concentration of DMSO used. 

AFP III supplemented with DMSO was tested on the cryopreservation of monolayer adenocarcinomic human alveolar basal epithelia (A549) cells [107]. The effects of “cryopreservation format” (Suspension Freezing vs. Monolayer Freezing) and cellular positioning (intracellular vs. extracellular) of the AFP were studied and findings were that extracellular AFP inclusion aided better monolayer cell recovery. Similar results were obtained when DMSO was combined with LeIBP in Mammalian cervical cancer cells (HeLa) cryopreservation [108]. In cryopreservation of human liver hepatocellular carcinoma (HepG2) cells, AFP III has been effective [109] while AFGPs from *Lolium perenne* glycoproteins were unproductive but the findings attempt to explain the mechanism of cellular ice crystal injury/damage [110].

In stem cells, Huelsz-Prince et al. [111] supplemented the hypothermic storage of mice intestinal organoids at varying developmental stages with 10 mg/mL AFGP at 4 °C for up to 120 h. Optimal hypothermic cell survival rates were obtained until 72 h of hypothermic storage which remained high after rewarming, especially for young and mature organoids. At 120 h, a decline in hypothermic survival was observed and after rewarming, none of the cells survived. These results suggest that the recovery rates of organoids after prolonged hypothermic storage (72 h) could depend on the developmental stage of the cells. It is also possible that improved cell survival following prolonged storage and rewarming is obtained if the method used by Kong et al. [112] (where cryoprotectant was also added to the rewarming solution to inhibit/reduce ice recrystallization) is applied in subsequent research on organoids cryopreservation. In another study, a combination of AFPs and antioxidants (hypotaurine and catalase) improved the post-thaw survival and viability of spermatogonial stem cells (SSCs) from *Ictalurus furcatus* (blue catfish) [113]. Testing this approach on human SSCs is essential because improved cryopreservation of SSCs would imply progress in reproductive medicine and present young males with effective options to preserve fertility prior to commencing chemotherapy, radiotherapy, immunotherapy and other treatments that would impact adversely on sperm production.

### 3.4. Pancreatic Cells (Insulinoma Cells)

A potent strategy in combating insulin dependent diabetes mellitus is the introduction of insulin-secreting cells into the hepatic portal vein. Unfortunately, such cells are not always readily available; therefore, an effective means of preserving them where insulin release is not lost has to be developed and standardized. Fish derived AFPs I, II, III and an AFGP have been used in a study [114] aimed at the hypothermic storage of rat insulinoma cells (RIN-5F cell-lines) at +4 °C where AFPs I and III were able to sustain the survival of 60% of the cells for up to 5 days. Additionally, an AFGP synthesized by Matsumoto et al. significantly reduced the nucleation and growth of ice, thereby enhancing the cryopreservation of rat islets [115]. 

The little evidence retrieved of AFP application in cryopreservation of pancreatic cells indicates the opportunity for more studies especially with human derived pancreatic cells. 

### 3.5. Tissue and Organs Cryopreservation 

#### 3.5.1. Ovaries and Ovarian Tissue (OT)

Human ovarian tissue (OT) cryopreservation is beneficial for the restoration of female reproductive capability following ovarian cancer chemotherapy. OT cryopreservation has been performed previously using DMSO [116], EG [117] and sucrose [118,119]. Nevertheless, these protocols have to be optimized as they are not entirely sufficient to avoid commonly observed downsides, such as damage to granulosa cells [120,121] and thecal layer thinning [122], which could lead to reduced follicle and tissue viability [122]. To investigate the application of AFPs in this field, Lee et al. [123] applied AFPs to the cryopreservation and transplantation of mouse ovarian tissue. Results showed this could prevent cryodamage to ovarian tissue, as seen from reduced follicular apoptosis, reduced plasma FSH after transplantation and better tissue density among other positive outcomes obtained. 

A combination of 5% DMSO and 0.1 mg/mL LeIBP has been beneficial in the cryopreservation of hamster ovary cells (CHO-K1) (99). The synergistic effect FfIBP and type III AFP at 10 mg/mL on the vitrification of mouse ovaries was carried out by Kim et al. in 2017 (17). Although the combination offered marked protection of ovarian follicles against cryodamage, the results were not statistically significantly different from when the AFPs were used individually. This suggests the need for more investigations into the synergistic effect of other AFPs at varying concentrations and storage conditions. Moreover, investigation into the clinical application of these findings and further studies to determine the exact mechanism through which IBPs are able to preserve ovarian tissue during the freeze-thaw cycle is recommended. 

Another approach is the addition of AFPs to the warming stage after vitrification with other cryoprotectants—a strategy tested and proven effective in the vitrification and warming of tissues [112,124]. Kong et al. [112] presented this evidence first in a study aimed at assessing the effect of adding an AFP (LeIBP) only during the warming phase of mouse OT vitrification. The outcome confirmed the relevance of inhibiting ice recrystallization inhibition and as well revealed the need to revise vitrification/warming protocols, as results obtained from adding AFPs to both vitrification and warming solutions versus their inclusion in warming solutions only were similar. 

In 2021, Kong et al. went further to investigate the effect of LeIBP in inhibiting IR during the warming of vitrified bovine ovarian tissue, which is closely anatomically related to the human OT. Findings revealed reduced changes and apoptosis in follicles, reduced fibrosis and better tissue micro-vascularization. Although the concentrations of AFP used may not have affected the outcome as results did not vary remarkably, 20 mg/mL of LeIBP showed high serum estradiol concentrations indicative of successful OT xenotransplantation post-cryopreservation [124]. 

A combination of 5% DMSO and 0.1 mg/mL LeIBP has been beneficial in the cryopreservation of hamster ovary cells (CHO-K1) [108]. The synergistic effect FfIBP and type III AFP at 10 mg/mL on the vitrification of mouse ovaries was carried out by Kim et al. in 2017 [25]. Although the combination offered marked protection of ovarian follicles against cryodamage, the results were not statistically significantly different from when the AFPs were used individually. This suggests the need for more investigations into the synergistic effect of other AFPs at varying concentrations and storage conditions. Moreover, investigation into the clinical application of these findings and further studies to determine the exact mechanism through which IBPs are able to preserve ovarian tissue during the freeze–thaw cycle is recommended. 

#### 3.5.2. Skin Grafts

Ibrahim et al. [125] employed Afp1m, an α-helix antifreeze peptide fragment occurring naturally in *Glaciozyma antractica* yeast in the cryopreservation of rat skin grafts and compared its activity with that of glycerol. Subzero storage was done at −4 °C for 72 h followed by 21 days post-transplantation examinations of the skin layers for variations in anatomy and mechanical properties. Results showed that 5 mg/mL of Afp1m is a prospective candidate as it appreciably preserved the skin grafts. In 2019, Khan et al. [126] conducted a similar study with grafts from Sprague Dawley rats. The skin grafts were subjected to subzero freezing at different temperatures and AFP concentrations and histological examinations showed acceptable cryopreservative ability of 5 and 10 mg/mL of Afp1m at −10 °C.

#### 3.5.3. Mammalian Hearts

Using the rat heart model, hypothermic preservation at 4 °C with marked conservation of hemodynamic and metabolic properties of the organ using 500 μg/mL of codfish AFGP for up to 120 min [127] and survival after subzero freezing with Arctic fish AFP III and AFP I have been documented [128,129] and attributed to the ability of AFPs to inhibit freezing, preserve hemodynamics and protect myocytes against apoptosis [130].

#### 3.5.4. Pancreatic Islet Grafts

Transplantation of Pancreatic Islet in the treatment of diabetes is often hindered by organ rejection via auto and/or alloimmune responses. Immunosuppressants are often administered to inhibit these immune responses but adversely, they can be toxic to the transplanted organs. Boris et al. [131] added a mimetic AFP—antiaging glycoprotein (AAGP) to human islets cultured in the presence of an immunosuppressant (tacrolimus) which resulted in better islet quality and yield, retained insulin secretion, less oxidative stress and reduced immune/inflammatory responses (release of interleukins 1b and 6, KC and TNF-a). This points to the possibility that AFPs can serve the dual purpose of protecting islet cells from immunosuppressant induced toxicity and ice formation induced damage during cryopreservation. A recent study by Dolezalova et al. [132] presents a unique approach to determining the diffusion kinetics of small molecules into human islets and improving cryopreservation using combined CPAs which was achieved by accelerating the penetration of trehalose into the islet core through incubation at 37 °C before the addition of DMSO. Exploring this technique with other impermeable CPAs can pave way for researchers to investigate the cryoprotective role of AFPs in a more efficient way and possibly extend this method to the cryopreservation of other tissues and organs such as bones [133], brain tissue [134], kidney [135] and lungs [136]. 

### 3.6. Food, Agriculture and Related Applications

The safety and quality of food are of principal consideration during storage; therefore, agents used in preservation have to be carefully selected to avoid the risk of toxicity/poisoning. An Ice structuring protein (ISP) isolated from wheatgrass juice was found to improve the freezing and thawing of commercially available pasta sauces [137]. More than 20% reduction in freezing and thawing time was observed which can substantially reduce processing time during freezing. The research, however, noted the high cost of obtaining ISPs, which is indicative of the need for cost effective/affordable AFPs. 

To confirm the effectiveness of this technique, further research into the preservation of flavour, colour, texture and nutritional value following freezing and thawing would be key. Notably, this is seen in studies to evaluate the effect AFP from *Hippophae rhamnoides* (seabuckthorn) seeds on green beans [138] and that of synthesized antifreeze peptides on frozen carrot [139] and cherries [140]. These studies revealed the ability of the proteins to preserve food texture, colour and prevent the escape of volatile constituents through cell membranes in addition to reducing freeze–thaw time. Oxidation occurring during the freezing of mirror carp (*Cyprinus carpio* L.) would usually result in loss of freshness by causing aggregation of microfibrillar proteins and loss of moisture. According to Du et al., the cryoprotective effect of ISP from ryegrass is a potential candidate for mirror carp storage as it minimized the damage [141]. 

Good quality bread has been produced from dough preserved with recombinant carrot antifreeze protein (132) and barley antifreeze protein [142]. In a later study, Liu et al. use frozen hydrated gluten to explain the cryoprotective mechanism of AFP in dough which includes protection of disulfide bonds, structural integrity and better rheological properties [143]. Other applications of AFPs in this area include carrot AFP in white salted noodles [144], Collagen AFP in silver carp fish surimi [145], *Tenebrio molitor* larvae AFP in vegetables [47], recombinant LeIBP in Korean beef [146], cold-acclimated oat AFP in ice cream [50], Herring AFP I in largemouth bass (*Micropterus salmoides*) [147] and red sea bream (*Pagrosomus major*) [148], where the combination with magnetic chitosan nanoparticles in both cases was also effective.

Fish AFPs I and III have been used in an attempt to stimulate the germination of tomato seeds which would generally not happen at temperatures below 10 °C [149]. Compared with the optimal conditions, where germination started on day 5 at 25 °C, treatment with AFP I was able to induce germination on day 16 at 20 °C but at a notably higher rate and larger seedling size with the best results obtained from the 100 µg/l AFP I treatment. A possible explanation is a reduced expression of catalase 1 (CAT1), H+ -ATPase and superoxide dismutase (SOD) germination genes, which was similarly observed in the inclusion of AFP to plant vitrification solution (PVS) in the cryopreservation of *Hosta capitata* meristems [150]. 

Alternatively, Seo et al. attribute the cryopreservative effect of AFP III on potato (*Solanum tuberosum*) shoot tips to upregulation of low temperature-related genes [151]. This infers that more investigations are required to provide clarity on the exact gene-related mechanism of cold tolerance conferred on the plants by AFPs and the probable inclusion of other assays to confirm cryoprotection like enthalpy (thermal hysteresis) analysis used by Jeon et al. to investigate the promotion of survival and germination of *Chrysanthemum morifolium* (Borami) shoots by AFP III [152]. Balamurugan et al. induced better cold-tolerant properties in tomatoes by expressing *Lolium perenne* AFP genes in transgenic tomato cell lines [153]. This is an innovative approach that would enable farmers skip the AFP treatment phase in seedlings germination. Nevertheless, it can only be approved for practical use after genetically modified organisms (GMO) concerns have been addressed.

## 4. Future Research Direction and Concluding Remarks 

Although the clinical application of AFPs is still limited, with some, e.g., human sperm [95] and blood [154] being tested, cryopreservation with AFPs have other unpopular applications, meaning that researchers are digging more into the potentials and novelty of AFP and broadening the scope of its application. In probiotic cryopreservation, Chen et al. used antifreeze peptides derived from tilapia scales [155] and a glycopeptide analogue [156] to successfully reduce oxidation, membrane rupture and other damaging effects of ice on *Streptococcus thermophilus*. 

LeIBP has also aided cryopreservation of cells like fibroblasts, proteoblasts and keratinocytes [108] and red blood cells [154]. AFP from *D. canadensis* shortened the onset of ice nucleation and produced better post thaw viability of embryonic rat thoracic aorta smooth muscle cells when used with DMSO [157]. Moreover, cryopreservation of marine microalgae using LeIBP and FfIBP could be very useful as hatchery feed in aquaculture and in by extension be applicable in biopreservation of scarce species to avoid extinction [158].

The AFP discovery and research trend (Figure 1) and Table 1 confirm that AFP is gaining more popularity as the chemical and molecular composition of the majority of the AFPs have been determined. Nonetheless, isolation, purification and accurate classification of AFPs remain challenging. In attempts to modify AFPs, research has shown that increasing the structural flexibility of AFPs would result in decreased binding to ice surfaces [159]. The large disparity between AFPs in sequence and structure also makes it a challenge to findi a reliable prediction model for their classification [160]. 

It is therefore pertinent to conduct studies focused on structural elucidation, classification and THA determination. Usman et al. [161] propose a creative machine learning method to predict and classify AFP sequences based on latent space encoding (LSE) resulting in increased accuracy over already existing algorithms. More of such innovative research designs would allow for the discovery of techniques required for modification, functional adjustment and mass production of AFPs.

The extensive clinical application of AFPs has also been hindered by toxicity and immunogenic responses [7] as seen in the diminished survival of human embryonic liver and kidney cells after AFP treatment [162,163]. Another problem is the formation of sharp bipyramidal extracellular ice when a high AFP concentration is applied [115], which would be detrimental during cryopreservation via vitrification. 

To overcome these obstacles, other methods are emerging. These methods include but are not limited to the use of AFP mimetics like nanoparticles [164] and synthetic polymers (e.g., polyvinyl alcohol) [165], synthesis of the antifreeze peptides [115,139], use of recombinant AFPs [166,167], the combination of AFP with other cryoprotectants such as magnetic nanoparticles [147,148], 2,3-Butanediol [168], glycerol [154] and a mixture of AFP isoforms [109]. Synthesizing antifreeze peptide analogues as a potential solution is faced with the difficulty of degradation by proteolytic mechanisms [169]. Therefore, techniques to optimize protein stability and achieve high-level expression is required.

AFP mimetics are materials that confer protection against cryoinjury by imitating the mechanism of action of AFPs and recently, many researchers are investigating these mimetics as a potential replacement for AFPs in the future. Polyvinyl alcohol, metallohelices and polyampholytes mimic AFPs by inhibiting ice recrystallization while ammonium polyacrylate and complexes of zirconium acetate act by inducing thermal hysteresis and shaping ice, respectively. From the understanding of the stages in cryopreservation, it is possible that the use of an AFP mimetic with a singular mechanism of action might not be completely effective throughout the cryopreservation process. Hence, we propose that supplementation with other mimetics of varying mechanisms might yield better results.

In the case an AFP’s use is only limited due to immunogenic reactions, it is suggested to research techniques to ensure complete AFP removal post-thaw and also to evaluate the effect of co-administering immunosuppressants. Another potent strategy to overcoming AFP toxicity is to study how they are metabolized and eliminated naturally in AFP utilizing organisms. This might provide insights on how to either simulate this process or restrict AFP use to only systems that are capable of such metabolic reactions. More research on the optimal conditions suitable to avert denaturing of AFPs is also encouraged.

This review has also revealed significant progress and potential commercialization of AFP utilization in food cryopreservation. This is probably because most of these food sources especially fruits and vegetables may share genealogy with naturally AFP producing plants hence better tolerance and cryopreservation outcome. Notwithstanding, in-depth safety and toxicity studies have to be conducted before such preserved foods can be consumed. 

The concept of cryopreservation is applicable in other atypical research fields such as spaceflight experiments involving living samples [170], cryosleep and cryo-hibernation [171], cryopreservation of animals and cryonics [172]. AFPs may also find suitable applications in optimizing cryopreservation procedures in the aforementioned areas.

In conclusion, AFP application in cryopreservation has shown good prospects for future optimization, validation, standardization and large-scale application. There is scientific evidence to support the successful application of AFP III in cryopreservation of human sperm, A549 cells, and HepG2 cells, and AAGP in cryopreservation of human islets. The few cases where indifferent or negative results were obtained for instance the unproductive use of Lolium perenne AFGPs in cryopreservation of HepG2 cells should be considered as pointers to where improvement and further research is needed rather than deterrents. AFP mimetics are gaining enormous research attention, we envision them as the solution to most of the challenges encountered in AFP applications and encourage studies in this direction. This and other novel investigations would permit the use of AFPs and its derivatives as potential ingredients in clinically approved cryopreservation solutions. 

## Figures and Tables

**Figure 1 ijms-23-02639-f001:**
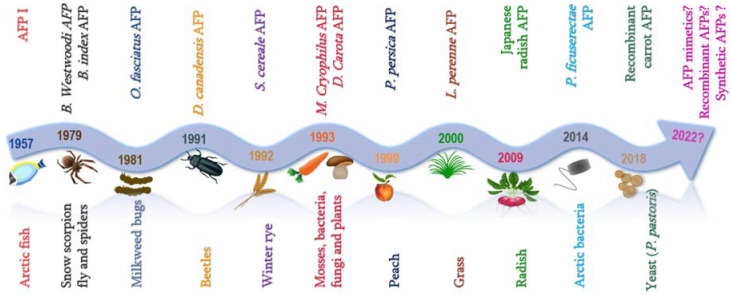
Trend in the discovery of Antifreeze proteins from diverse biological sources.

**Figure 2 ijms-23-02639-f002:**
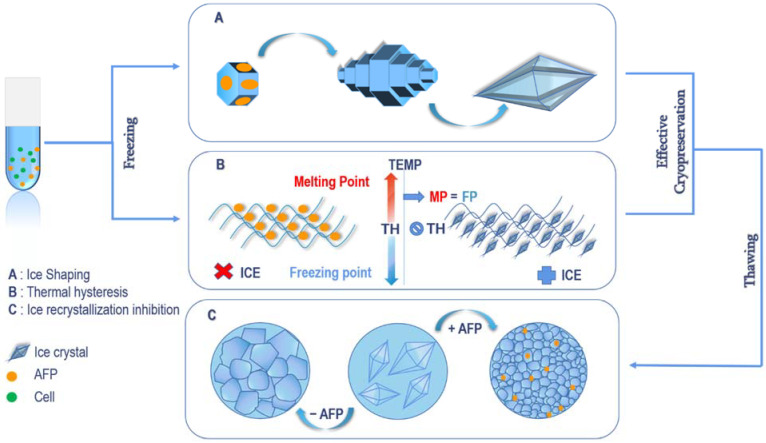
Mechanism of action of AFPs.

**Table 1 ijms-23-02639-t001:** Source and chemistry of some AFPs.

	Origin/Source	Chemical Structure	Size (kDa)	TH Value	THA (Reference)
FfIBP	*Flavobacterium frigoris* PS1	β-Helix	~25.3	2.2 °C at 0.13 mg/mL	Hyperactive [24]
Type III AFP	Ocean pout, wolffish, eelpout, *Zoarces elongatus* Kner	Globular	6.5	0.23 °C at 1 mg/mL	Moderate [25,26]
LeIBP	Arctic yeast (*Leucosporidium* sp.)	Dimeric righ-handed β-Helix fold	~25	0.17 °C at 50 µM or 0.34 °C at 10.8 mg/mL	Moderate [24,27,28]
TaAFP	Wheat bran (*Triticum aestivum*)	β-sheet structures	3.63		[29,30]
DcAFP	Carrot (*Daucus carota*)	ND	36	0.35 °C at 1 mg/mL	Low [26]
EfcIBP	Antarctic ciliate (*Euplotes focardii*) associated bacterium	β-sheet structures	~25	0.53 °C at 50 µmL	Moderate [31,32]
TisAFP8/6	Snow mould fungus (*Typhula ishikariensis*)	Right-handed β-helix with a long α-helix insertion	~23	TisAFP8: 2.0 °C at 0.11 mM TisAFP6: ~0.3 °C at 0.11 mM	TisAFP8: Hyperactive, TisAFP6: Moderate [33,34,35]
DAFP	*Dendroides canadensis*	Mostly β-sheet structures	~8.7 to 7.4	~4 °C at ~0.5 mg/mL	Hyperactive [36]
ColAFP	*Colwellia* sp. strain SLW05	Irregular β-helical structure	~25	~4 °C at 0.14 mM	Hyperactive [37,38]
RiAFP	Longhorn beetle (*Ragium inquisitor*)	β-strands	12.8	~6.3 °C at 0.075 mM	Hyperactive [39]
Type II AFP	Atlantic herring (*Clupea harengus*) and smelt fish, sea raven (cottid)	β-strands stabilized by di-sulfide bridge	92.99	ND	Moderate [40]
LpAFP	Ryegrass (*Lolium perenne*)	β-helix	29	0.1 °C in water at 5 mg/mL	Low [41,42]
shsAFP	Shorthorn sculpin (*Myoxocephalus scorpius*)	α-helix	3.780	~0.75 °C at 8 mg/mL	ND [43]
ApAFP752	Desert beetle (*Anatolica polita*)	Right-handed β-helix	30	0.45 °C at 0.5 mg/mL	ND [44]
SM-AFP	Silkworm cocoon		1.00950	0.94 °C at 5 mg/mL	[45]
TmAFP	Mealworm beetle (*Tenebrio molitor*)	Right-handed β-helix	8.4	5.5 °C at 1 mg/mL	Hyperactive [37,46,47,48]
BaAFP-1	Cold acclimated malting barley (*Hordeum vulgare*)	ND	13.18	1.04 °C at 18.0 mg/mL.	ND [49]
AsAFP	Cold-Acclimated Oat (*Avena sativa*)	ND	~22.08	1.24 °C at 50 mM,	ND [50]
MpAFP	Antarctic bacterium (*Marinomonas primoryensis*)	Right-handed long β -helical fold	40	2.0 °C at 0.5 mg/mL	Hyperactive [37,51]
TYPE I AFP	Winter flounder (*Pseudopleuronectes americanus*), flatfish and sculpin	α-helical structure	6.48	0.27 °C at 400 µM	Moderate [37,40,52]
fcAFP	Polar diatom (*Fragilariopsis cylindrus*)	ND	25.939	0.9 °C at 350 μM	Moderate [53]
NagIBP	*Navicular glaciei*	ND	~25	3.22 °C at 1.6 mM.	Hyperactive [54,55]
AFP 4	Antarctic yeast (*Glaciozyma antarctica)*	β-helix	~25	0.08 °C at 200 μM	ND [56]
CfAFP, sbwAFP	Spruce budworm (*Choristoneura fumiferana*)	Regular left-handed β-helix	9	1.08 °C at 20 µM	Hyperactive [52,57,58]
AnAFP	*Ammopiptanthus nanus*	ND	119.24	0.46 °C at 20 mg/mL	ND [59]
AnAFP	Ammopiptanthus mongolicus	α-helix (11%), antiparallel β-sheet (34%)and random coil (55%)	37.100	0.35 °C at 5 mg/mL	ND [60]
Type IV AFP	Longhorn sculpin (Myoxocephalus octodecimspinosis)	Anti-parallel helical bundle structure	12.296	~0.5 °C at 2 mM	ND [61]
ND	Spruce needles (*Picea abies* and *Picea pungens*)	ND	7–80	2.0 °C at 400 μg/mL	ND [62]

ND: Not determined.

## Data Availability

All data used to support the findings of this review are included in the article.

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
