# Peer review of "Antifreeze Proteins: Novel Applications and Navigation towards Their Clinical Application in Cryobanking"

_ijms, 2022, doi:10.3390/ijms23052639_

Round 1

Reviewer 1 Report

The paper by Ekpo and co-authors entitled “Antifreeze Proteins: Novel Applications and Navigation To-2 wards Their Clinical Application in Cryobanking” reviewed the potential clinical application of antifreeze proteins. In general it is written in a clear and understandable language and style. I recommend this article for publication with the minor revisions suggested.

-Line 10: Correct sentence because the word “fish” is duplicated.

-Line 36: Change “these disadvantages is” to “these disadvantages are”.

-Figure 2 should be presented before section 2.

-Section 2 only presents Table 1 but without any discussion. Is there any coincidence in the different AFPs presented? The origin/chemical structure/size gives some relevant information from future studies? Which are the best AFPs based on the thermal hysteresis? Please, add a discussion in this section.

-Table 1: Use the same format in the table. For example: use “β sheet” or “β-sheet”, “mg/mL” or “mg mL-1”, not both terms. Control the use of italics and capital letters. I suggest the addition of “ND” (Not determined) in the missing information. What are the bases of the classification of TH activity?

-Line 97: Change “In vitro” to “in vitro”.

-Line 100: Surname should start with capital letter.

-Lines 100-101: Change “the effect AFPs” to “the effect of AFPs”.

-Lines 119-120: Scophthalmus maximus should be in italics.

-Line 147: Cyprinus carpio should be in italics.

-Line 156: Dot is missing after (91).

-Lines 176 and 250: Change “4oC”.

-Line 248: Change “-10oC”.

-Line 252: Change “mins” to “min”.

-Line 254: Dot is missing after (121).

-Line 289: Dot is missing after (133).

-Line 308: Solanum tuberosum should be in italics.

-Line 333: Streptococcus thermophiles should be in italics.

Reviewer 2 Report

The submitted review article provides an overview on the positive/negative effects of antifreeze proteins in the cryoconservation of cells and  tissues. In general the manuscript is well done and structured, but in my opinion it remains quite descriptive for the standard of the journal.  The novelty and the need of this kind of review'd be better explained. Similarly, par 3.2 and 4  could be improved. 

Minor points:

correct several typos

the name of species  always requires  Italic font

Reference list is not formatted accordingly to the journal's instructions. 

Not all abbreviations have been formatted at the first appearence in the main text.

Reviewer 3 Report

The manuscript-Antifreeze Proteins: Novel Applications and Navigation Towards Their Clinical Application in Cryo-banking- reviews the role of AFP in various applications related to health and food. 

Specific comments are given below:

Page 2, Line 46 - You categorize these CPA into permeable (DMSO, EG, PRoH) and non-permeable (Sucrose, trehalose etc) ones.

Figure 2 and Page 2 first section tries to explain the mechanism of action of AFP in brief. It is better to give some more recent citations on the mechanism of action of AFPs- 1) Flores A, Quon JC, Perez AF, Ba Y. Mechanisms of antifreeze proteins investigated via the site-directed spin labeling technique. Eur Biophys J. 2018;47(6):611-630. doi:10.1007/s00249-018-1285-3

2)Perez AF, Taing KR, Quon JC, Flores A, Ba Y. Effect of Type I Antifreeze Proteins on the Freezing and Melting Processes of Cryoprotective Solutions Studied by Site-Directed Spin Labeling Technique. Crystals (Basel). 2019;9(7):10.3390/cryst9070352

Page 6 section on Sperm- The recent work by Chen B et al hypothesized a novel molecular mechanism for cryoprotection that AFP III may reduce the release of cytochrome c and thereby reduce sperm apoptosis by modulating the production of ROS in mitochondria (Front. Endocrinol., 28 May 2021 | https://doi.org/10.3389/fendo.2021.672619) and the study by Zandiyeh S et al- on human sperm (https://doi.org/10.1016/j.repbio.2020.03.006) may also considered for this section

Section 3.3- Cancer cells....Better substitutes to DMSO- In cryopreservation it is always shown that the permeable CPAs are superior compared to trahalose or other carbohydrates whicha re Non-permeable. However a combination may reduce the toxicity of the permeable one. So you may say additives like trahalose or AFP...

Secion 3.3 last section on spermatogonial stem cells: You may expand little bit more on this -Cryopreservation of spermatogonial stem cells (SSCs) is an applicable method for young males seeking fertility preservation before starting a treatment for chemo/radio therapy/immunotherapy. 

section 3.4- You can include the study by Dolezalova N- developed a method to determine diffusion kinetics of small molecules in aqueous solutions into the core of pancreatic islets and to investigate strategies to accelerate it. https://www.nature.com/articles/s41598-021-89853-6 

The above study pave way to researchers in the field of AFP to investigate its role in a more efficient way

Conclusion and future direction: Diverse AFPs found in fishes (Type I, II, III, IV and antifreeze glycoproteins (AFGPs)), are sub-types and show low sequence and structural similarity, making their accurate prediction challenging. machine-learning methods have been proposed for the classification of AFPs, prediction methods that have greater reliability are being explored https://www.nature.com/articles/s41598-020-63259-2

Secondly the role of AFPS in Space research- CRyosleep and cryopreservation of living materials being transported to space etc.-DOI: 10.1016/j.biomaterials.2021.120673

Author Response

Response to reviewer 3 comments

The manuscript-Antifreeze Proteins: Novel Applications and Navigation Towards Their Clinical Application in Cryo-banking- reviews the role of AFP in various applications related to health and food. Specific comments are given below:

Point 1: Page 2, Line 46 - You categorize these CPA into permeable (DMSO, EG, PRoH) and non-permeable (Sucrose, trehalose etc) ones.

Response 1: Thanks for your kind comments. We have incorporated your suggestion as follows: To overcome these drawbacks, the addition of cryoprotectants or cryoprotective agents (CPAs) before sample freezing or vitrification has become the main stay [6]. CPAs are categorized as permeable (dimethyl sulfoxide (DMSO), proline, glycerol, ethylene glycol (EG), propylene glycol (PG), etc.) or impermeable (polyvinyl alcohol (PVA), polyampholytes, trehalose, sucrose, graphene oxide, etc.) based on whether they can cross cell membranes [7]”.

Point 2: Figure 2 and Page 2 first section tries to explain the mechanism of action of AFP in brief. It is better to give some more recent citations on the mechanism of action of AFPs- 1) Flores A, Quon JC, Perez AF, Ba Y. Mechanisms of antifreeze proteins investigated via the site-directed spin labeling technique. Eur Biophys J. 2018;47(6):611-630. doi:10.1007/s00249-018-1285-3 2) Perez AF, Taing KR, Quon JC, Flores A, Ba Y. Effect of Type I Antifreeze Proteins on the Freezing and Melting Processes of Cryoprotective Solutions Studied by Site-Directed Spin Labeling Technique. Crystals (Basel). 2019;9(7):10.3390/cryst9070352

Response 2: Thank you. We have incorporated your suggestion into the new manuscript: The interaction between AFPs and water and ice crystals at low temperatures to inhibit ice formation and growth has been studied and confirmed with diverse methodologies including quantum filtering-spin exchange nuclear magnetic resonance (NMR) [17] microfluidic experimentation [18], NMR microimaging and the more recent site-directed spin labeling performed by monitoring the activity of spin-labelled AFPs at low temperatures with electron paramagnetic resonance (EPR) and cryo-photo microscopy [19,20].”

Point 3: Page 6 section on Sperm- The recent work by Chen B et al hypothesized a novel molecular mechanism for cryoprotection that AFP III may reduce the release of cytochrome c and thereby reduce sperm apoptosis by modulating the production of ROS in mitochondria (Front. Endocrinol., 28 May 2021 | https://doi.org/10.3389/fendo.2021.672619) and the study by Zandiyeh S et al- on human sperm (https://doi.org/10.1016/j.repbio.2020.03.006) may also considered for this section.

Response 3: Thank you for the recommendation. We had previously included these studies in this section “AFPs including AFP I, AFP III, DAFP (Dendroides canadensis antifreeze protein) and AFGPs have been applied in sperm banking research e.g. sea bream sperm [85,86], buffalo bull (Bubalus bubalis) sperm [87-90], ram sperm [91], rabbit sperm [71], Persian sturgeon (Acipenser persicus) sperm [92], rooster sperm [93], common carp (Cyprinus carpio) sperm [94], human sperm [95] and crab-eating macaque (Macaca fascicularis) sperm [96] where later research has provided more insight into the possible mechanism of action of AFP III [97]. AFP III improved survival by preserving protein expression, and modulating the release of cytochrome C and free radicals at molecular level.”

Point 4: Section 3.3- Cancer cells....Better substitutes to DMSO- In cryopreservation it is always shown that the permeable CPAs are superior compared to trahalose or other carbohydrates whicha re Non-permeable. However a combination may reduce the toxicity of the permeable one. So you may say additives like trahalose or AFP...

Response 4: Thank you for your comment. We agree with your suggestion and we have restructured the sentence as follows:A potent approach to diminish the toxic effect of DMSO in the cryopreservation of cancer and stem cells has been a combination with other CPAs like trehalose [100], sucrose [101], polyampholytes [102] and AFPs. This creates a synergy in ice inhibiting mechanisms while reducing the total concentration of DMSO used.”

Point 5: Secion 3.3 last section on spermatogonial stem cells: You may expand little bit more on this -Cryopreservation of spermatogonial stem cells (SSCs) is an applicable method for young males seeking fertility preservation before starting a treatment for chemo/radio therapy/immunotherapy. 

Response 5: This is a very important point to consider and we have included it in the new manuscript “AFPs and antioxidants (hypotaurine and catalase) improved the post-thaw survival and viability of spermatogonial stem cells (SSCs) from Ictalurus furcatus (blue catfish) [109]. Testing this approach on human SSCs is essential because improved cryopreservation of SSCs would imply progress in reproductive medicine and present young males with effective options to preserve fertility prior to commencing chemotherapy, radiotherapy, immunotherapy and other treatments that impact adversely on sperm production.” Thank you

Point 6: section 3.4- You can include the study by Dolezalova N- developed a method to determine diffusion kinetics of small molecules in aqueous solutions into the core of pancreatic islets and to investigate strategies to accelerate it. https://www.nature.com/articles/s41598-021-89853-6. The above study pave way to researchers in the field of AFP to investigate its role in a more efficient way.

Response 6: We agree with your kind suggestion and we have added it to section 3.5.4 where we discuss pancreatic islets as follows: “A recent study by Dolezalova et al. [128] presents a unique approach to determining the diffusion kinetics of small molecules into human islets and improving cryopreservation using combined CPAs which was achieved by accelerating the penetration of trehalose into the islet core through incubation at 370C before the addition of DMSO. Exploring this technique with other impermeable CPAs can pave way for researchers to investigate the cryoprotective role of AFPs in a more efficient way and possibly extend this method to the cryopreservation of other tissues and organs like bones [129], brain tissue [130], kidney [131], lungs [132].”

Point 7: Conclusion and future direction: Diverse AFPs found in fishes (Type I, II, III, IV and antifreeze glycoproteins (AFGPs)), are sub-types and show low sequence and structural similarity, making their accurate prediction challenging. machine-learning methods have been proposed for the classification of AFPs, prediction methods that have greater reliability are being explored https://www.nature.com/articles/s41598-020-63259-2. Secondly the role of AFPS in Space research- CRyosleep and cryopreservation of living materials being transported to space etc.-DOI: 10.1016/j.biomaterials.2021.120673.

Response 7: Thank you for the insightful suggestion. We have adjusted the manuscript accordingly: Nonetheless, isolation, purification and accurate classification of AFPs remains challenging. In attempts to modify AFPs, research has shown that increasing structural flexibility of AFPs would result in decreased binding to ice surfaces [155]. The large disparity between AFPs in sequence and structure also possess a challenge to finding a reliable prediction model for their classification. It is therefore pertinent to conduct studies focused on structural elucidation, classification and THA determination. Usman et al. [156] proposes a creative machine-learning method to predict and classify AFP sequences based on latent space encoding (LSE) resulting in increased accuracy over already existing algorithms. More of such innovative research designs would allow for discovery of techniques required for modification, functional adjustment and mass production of AFPs.

AND

“The concept of cryopreservation is applicable in other atypical research fields such as spaceflight experiments involving living samples [165], cryosleep and cryo-hibernation [166], cryopreservation of animals and cryonics [167]. AFPs may also find suitable application in optimizing cryopreservation procedures in the afore mentioned areas”.

Finally, we appreciate your expert comments, observations and the time taken to provide us with highly relevant recent study references. This has immensely improved on the quality of our manuscript.

Reviewer 4 Report

This review highlights the main applications of antifreeze proteins from various sources, and their potential applications in the future. Overall, this review provides a good overview of the current applications of antifreeze protein, however, the information presented should be elaborated upon and additional information should be added before it can be considered for publication.

For example, this review does not cover or mention the processes or degradation mechanisms that AFP/AFGPs undergo, which makes them often unsuitable for many applications. Before a discussion on applications of AFPs can be presented, these issues must be presented and discussed, as AFP degradation is a major limiting factor that limits where they can be implemented. Please include a discussion on these, and cite current literature that discusses the issues with AFP denaturing and degradation, and how it is currently being addressed.

In addition, there is currently extensive research being conducted on antifreeze protein mimics, which are intended to be used as substitutes for various AFP/AFGPs, due to the delicate nature of many AFP/AFGPs. For example, PVA is currently being considered as a substitute for AFP/AFGPs, and has shown considerable promise, when modeled off of AFP/AFGPs. This should be considered and mentioned as well.

The authors also could elaborate on their conclusions of the review, in which areas they think research should be focus. Currently, the authors only mention current research directions, but do not give their own personal view of the topic, and areas that could be addressed, gaps in current research, and oversights that should be remedied. All of these should be included in the discussion section.

Reviewer 5 Report

The review focuses mainly on clinical applications of AFP, as stated in the title. However, I see a lack of the AFPs biochemical properties description. In the review is only reported a drowing that summirize the activity, but in my opinion the functional properties should be more detailed. Some ideas may derive from the following paper (already cited in the review): 10.1111/febs.13965.

Minor modifications: 

- in table I: I would change Euplotes focardii  into Euplotes focardii (Antarctic ciliate) associated bacterium including the following reference that well explain the identification of the AFP: 10.1017/S0954102014000017

Round 2

Reviewer 3 Report

Thank you for incorporating the suggestions to make it more comprehensive. This review will be a useful addition to literature for researchers working on AFP and Cryopreservation.

Reviewer 4 Report

The issues have been addressed. The paper can be accepted in the present form.

Author Response

Thanks a lot